



# Assessment of errors and biases in retrievals of $X_{CO2}$, $X_{CH4}$, $X_{CO}$, and $X_{N2O}$ from a 0.5 cm$^{-1}$ resolution solar viewing spectrometer

Jacob K. Hedelius[1], Camille Viatte[1], Debra Wunch[1,a], Coleen Roehl[1], Geoffrey C. Toon[2,1], Jia Chen[3,b], Taylor Jones[3], Steven C. Wofsy[3], Jonathan E. Franklin[4,c], Harrison Parker[5], Manvendra K. Dubey[5] and Paul O. Wennberg[1]

[1]California Institute of Technology, Pasadena, CA, USA
[2]Jet Propulsion Laboratory, California Institute of Technology, Pasadena, CA, USA
[3]Harvard University, Cambridge, MA, USA
[4]Dalhousie University, Halifax, Nova Scotia, Canada
[5]Los Alamos National Laboratory, Los Alamos, NM, USA
[a]now at University of Toronto, Toronto, Ontario, Canada
[b]now at Technische Universität München, Munich, Germany
[c]now at 3

*Correspondence to:* J. Hedelius (jhedeliu@caltech.edu)

**Abstract.** Bruker$^{TM}$ EM27/SUN instruments are mobile commercial solar-viewing near-IR spectrometers. They show promise for expanding the global density of atmospheric column measurements of greenhouse gases and are being marketed for such applications. They have been shown to measure the same variations of atmospheric gases within a day as the high resolution spectrometers of the Total Carbon Column Observing Network (TCCON). However, there is little known about the long-term stability and uncertainty budgets of EM27/SUN measurements. In this study, which includes a comparison of 186 measurement days spanning 11 months, we note that atmospheric variations of $X_{gas}$ within a single day are well captured by these low-resolution instruments, but over several months, the measurements drift noticeably. We present comparisons between EM27/SUN instruments and the TCCON using GFIT as the retrieval algorithm. In addition, we perform several tests to evaluate the robustness of the performance and determine the largest sources of errors from these spectrometers. We include comparisons of $X_{CO2}$ and $X_{CH4}$, and for the first time present comparisons of $X_{CO}$ and $X_{N2O}$. Specifically we note EM27/SUN biases of 0.03%,



0.75%, -0.12%, and 2.43% for $X_{CO2}$, $X_{CH4}$, $X_{CO}$, and $X_{N2O}$ respectively with $1\sigma$ precisions of 0.08% and 0.06% for $X_{CO2}$ and $X_{CH4}$ from measurements in Pasadena. We also identify significant error caused by non-linear sensitivity when using an extended spectral range detector used to measure CO and $N_2O$.

# 1 Introduction

Measurements of atmospheric mixing ratios of greenhouse gases (GHG), including $CO_2$ and $CH_4$, are needed to aid in estimating fluxes, flux changes, and to ensure international treaties to reduce emissions are fulfilled. The Total Carbon Column Observing Network (TCCON) makes daytime column measurements of these gases. The Orbiting Carbon Observatory-2 (OCO-2) and GOSAT missions enable column GHG measurements with global coverage. These GHG monitoring satellites make

measurements at one time of day and, therefore, lack the temporal resolution that a dedicated ground site provides.

Due to cost, lack of infrastructure, and stringent network requirements, there are limited ground sites on a global scale, e.g. there are no TCCON sites currently in operation in either Africa, South America or central Asia (Wunch et al., 2015), and there currently is no urban area with more than one TCCON site.

Cheaper, portable, solar-viewing Fourier transform-spectrometers (FTS) can make contributions in these settings provided they have long-term stability. The Bruker Optics[TM] EM27/SUN, with the "SUN" indicating a built-in solar tracker, is a transportable FTS that may supplement global GHG measurements made by current networks (Gisi et al., 2012). This unit is small and stable enough to easily be transported for field campaign measurements, including measurements at multiple locations in

one day. Column averaged dry air mole fractions (DMF) of gases ($X_{gas}$) are retrieved from the EM27/SUN measurement, like the TCCON. Retrieved $X_{gas}$ has been compared with a co-located TCCON site in Karlsruhe, Germany in past work for 26 days of $X_{CO2}$ retrievals from one EM27/SUN instrument (Gisi et al., 2012), and 6 days of both $X_{CO2}$ and $X_{CH4}$ retrievals from five EM27/SUN instruments (Frey et al., 2015).

Operators of these instruments have different end goals to better understand the carbon cycle. $X_{CO2}$ and $X_{CH4}$ retrievals from these instruments have been compared with satellite measurements in areas without a TCCON site (Klappenbach et al., 2015) as well as with satellite measurements in highly



polluted areas (Shiomi et al., 2015). Emission flux estimates from the Berlin area ($<30\times30$ km$^2$) were made by combining upwind/downwind measurements from five spectrometers and were compared with a simulation (Hase et al., 2015). Chen et al. (2016) have assessed gradient strengths around a large dairy farm (~100,000 cows) in Chino, California ($<12\times12$ km$^2$) using measurements from upwind/downwind

spectrometers. Weather Research and Forecast Large-eddy Simulations (WRF-LES, 4 km resolution) were used in combination with 4 simultaneous measurements to estimate fluxes from specific grid boxes in a sub-region of the Chino dairy farm area which is within a larger urban area (Viatte et al., 2016).

The column measurements used in these studies provide some advantages over in situ measurements

including less sensitivity to vertical exchange, surface dynamics, and small scale emissions (McKain et al., 2012), which are difficult to model. Though column measurements can depend on mixed layer (ML) height in highly polluted areas, generally column measurements depend primarily on regional-scale meteorology, and regional fluxes (Wunch et al., 2011b; McKain et al., 2012). For example, Lindenmaier et al. (2014) used observations from a single TCCON site to verify one day of emissions

from coal power plants of about 2000 MW each at ~4 km and 12 km away. Because of their large spatial sensitivity, column measurements are well suited for estimation of net emissions, model comparison, and satellite validation. A single site was used to estimate Los Angeles, California (L.A.) emissions based on a sufficiently accurate emissions inventory and the observation that $X_{gas}$ anomalies within L.A. are highly correlated (Wunch et al., 2009). Generally though, a single column measurement

site is insufficient to estimate emissions from an entire urban region (Kort et al., 2013). However, multiple column measurements can be combined to characterize part or all of an urban area (Hase et al., 2015; Chen et al., 2016;Viatte et al. 2016)

The main goal of this work is to quantitatively evaluate the robustness of EM27/SUN retrievals long-term. This is accomplished by comparing retrievals from the EM27/SUN with a co-located standard

(TCCON site) at Caltech, in Pasadena, California, United States. TCCON spectrometers make the same type of measurements (direct solar near-infrared) at high spectral resolution. Here we report $X_{CO2}$, $X_{CH4}$, $X_{CO}$ and $X_{N2O}$ comparison measurements from an EM27/SUN. The $X_{CO}$ and $X_{N2O}$ measurements were made possible by a detector with an extended spectral range provided by Bruker$^{TM}$. The EM27/SUN



$X_{CO2}$ and $X_{CH4}$ to TCCON comparison is the longest to date; 186 measurement days spanning 11 months. In part of January 2015 an additional 3 EM27/SUN instruments were at Caltech for 9 to 12 days of $X_{CO2}$ and $X_{CH4}$ comparisons to assess their relative biases. In Section 2 we briefly describe differences in instruments and data acquisition. In Section 3 we describe the retrieval software. In Section 4 we describe inherent properties of EM27/SUNs such as instrument line shapes (ILS), frequency shifts, ghosts, detector linearity, and external mirror degradation. Section 5 focuses on biases and sounding precision of different gases compared with the TCCON. Section 6 describes sources of instrumental error. We conclude with general recommendations of tests to perform on any new type of direct solar near-infrared (IR) instrument used to retrieve abundances of atmospheric constituents.

## 2 Instrumentation

### 2.1 TCCON 125 HR spectrometer

All TCCON sites employ the high-resolution Bruker Optics$^{TM}$ 125HR spectrometer that has been described in detail elsewhere (Washenfelder et al., 2006; Wunch et al., 2011b). For the Caltech TCCON site (34.1362°N, 118.1269°W, 237 m.a.s.l.), the 125HR uses an extended InGaAs detector covering 3800–11000 cm$^{-1}$ for detection and retrieval of all gases relevant to this study ($O_2$, $CO_2$, $CH_4$, CO, and $N_2O$). Figure 1 has example spectra from a 125 HR spectrometer and EM27/SUN instruments with the regions where individual gases are retrieved highlighted. Oxygen ($O_2$) abundance is useful in calculating the DMF because it represents the column of dry air and is combined with the column of the gas of interest to yield the DMF (Wunch et al., 2010)

$$X_{gas}=0.2095 \frac{\text{column}_{gas}}{\text{column}_{O_2}}$$ 

(1)

The Caltech 125HR spectrometer uses a resolution of approximately 0.02 cm$^{-1}$ (with a maximum optical path difference (MOPD) of 45 cm). It takes about 170 s to complete one forward/backward scan pair. TCCON sites have single sounding $2\sigma$ uncertainties of 0.8 ppm ($X_{CO2}$), 7 ppb ($X_{CH4}$), 4 ppb ($X_{CO}$), and 3 ppb ($X_{N2O}$) (Wunch et al., 2010). TCCON data are tied to the World Meteorological Organization (WMO) in situ trace gas measurement scale through extensive comparisons with in situ profiles obtained from aircraft and balloon flights. We use the TCCON as a standard against which to compare



the EM27/SUN instruments. TCCON data from this study are publically available from the Carbon Dioxide Information Analysis Center (Wennberg et al., 2014).

## 2.2 Caltech EM27/SUN

EM27/SUN spectrometers have been described elsewhere (Gisi et al., 2012; Frey et al., 2015; Klappenbach et al., 2015) so we focus on differences in setup and acquisition here. Most use the standard InGaAs detector sensitive to the spectral range spanning 5500–12000 cm$^{-1}$ which permits detection of $O_2$, $CO_2$, $CH_4$, and $H_2O$ (Frey et al., 2015). For this study, the Caltech EM27/SUN was delivered with an extended-band InGaAs detector sensitive to 4000–12000 cm$^{-1}$ which allowed for additional measurements of CO and $N_2O$ (Fig. 1). All EM27/SUN spectrometers used here had the typical MOPD of 1.8 cm, corresponding to a spectral resolution of 0.5 cm$^{-1}$. Interferograms (ifgs) were acquired in DC coupled mode to allow post-acquisition low-pass filtering of brightness fluctuations to reduce the impact of variable aerosol and cloud cover effects (Keppel-Aleks et al., 2007). Ghosts were reduced as data were acquired by employing the interpolated sampling option provided by Bruker$^{TM}$ (see also §4.3). A 10 KHz laser fringe rate is used to reduce scanner velocity deviations and each forward/backward scan took 11.6 s, or 5.8 s per individual measurement. To be more consistent with the TCCON measurements, no spectrum averaging or interferogram apodization was applied before retrieving DMFs. We recommend averaging only after retrievals if disc storage and processor speeds are sufficient, so spurious data can be filtered. To test the pre versus post averaging effect we used 9 retrieval days with 26,000 forward/backward measurements and used Bruker$^{TM}$ OPUS software to create spectra from ifgs. We compared retrievals from using 5 combined backward/forward measurements averaged pre with those averaged post retrieval. We also compared combined forward/backward measurements using a Norton-Beer apodization with those using no apodization. Results are in Table 1 and suggest that different averaging methods cause only small inconsistencies, under ~0.02% for $X_{CO2}$ and $X_{CH4}$.

The EM27/SUN was placed within 5 m of the Caltech TCCON solar tracker mirrors on the roof of the Linde+Robinson building (Hale, 1935). Measurements started on 2 June 2014 and are ongoing, but for this study, we include 186 measurement days that end on 4 May 2015. About 800,000 individual



EM27/SUN measurements and 40,000 individual TCCON measurements were acquired over this period. Of these about 580,000 and 15,000 were considered coincident and were not screened out by our quality control filters. After averaging data into 10 minute bins, there were about 6,000 comparison points.

## 2.3 LANL and Harvard EM27/SUN instruments

Three additional EM27/SUN instruments were compared with the Caltech TCCON site in January 2015—one owned by Los Alamos National Laboratory (LANL) and two owned by Harvard University (HU). To be consistent, all the acquisition and retrieval settings were the same as for the Caltech EM27/SUN. As opposed to the Caltech EM27/SUN, the LANL and HU instruments used the original InGaAs detector type sensitive over 5500–12000 cm$^{-1}$ (Frey et al., 2015). The LANL instrument, however, has a different high pass filter allowing it to measure up to 14500 cm$^{-1}$. This different filter is neither beneficial nor disadvantageous to this instrument as no gas column amounts are retrieved in that region. The LANL instrument was first used in January 2014 and has been compared with multiple TCCON sites in the U.S.A. including sites at Four Corners, LANL, NASA Armstrong, Lamont, Park Falls, and multiple Caltech comparisons (Parker et al., 2016). The HU instruments have been operational since May 2014 and were compared against each other at Harvard before traveling over 4100 km to Caltech. As noted by Gisi et al. (2012) and Chen et al. (2016), the ILS of these instruments is remarkably stable considering the long distances they travelled.

## 3 Retrieval software

SFIT (Pougatchev et al., 1995), PROFFIT ("PROFile fit", Hase et al., 2004), and GFIT (Wunch et al., 2015) are the three widely used retrieval algorithms to fit direct solar spectra and obtain column abundances of atmospheric gases. PROFFIT is maintained by the Karlsruhe Institute of Technology (KIT) and has been used to obtain DMFs from EM27/SUN instruments as well as NDACC-IRWG sites (Gisi et al., 2012; Frey et al., 2015; Hase et al., 2015). GFIT is maintained by the Jet Propulsion Laboratory (JPL) and has been used to obtain DMFs from other low resolution instrument measurements (e.g. an IFS 66, see Petri et al., 2012), in addition to being used to retrieve DMFs from





the MkIV spectrometer in balloon-borne measurements (Toon, 1991) and for the Atmospheric Trace Molecule Spectroscopy Experiment (ATMOS) flown on the space shuttle (Irion et al., 2002). GFIT is the retrieval algorithm of the TCCON (Wunch et al., 2011b). We chose to use GFIT for our analysis because 1) we want to be consistent with the TCCON for comparison and 2) the GGG software suite

containing GFIT is open-source allowing us to adapt routines if needed. We used the GGG2014 version for retrievals (Wunch et al., 2015).

All retrievals used the same pTz and $H_2O$ modeled profiles as well as the same a priori profiles (Wunch et al., 2015). We also used the same meteorological surface data for retrievals from all five instruments. All retrievals also used the same 0.2 hPa surface pressure offset. This offset was determined by

comparing measurements from the standard barometer with a calibrated Paroscientific Inc. 765-16B Barometric Pressure Standard that has a stated accuracy of better than 0.1 hPa.

### 3.1 I2S-double Sided

TCCON uses the interferogram-to-spectrum (I2S) subroutine part of GGG to perform fast Fourier transforms (FFTs) to create spectra from ifgs (Wunch et al., 2015). Though the Bruker[TM] OPUS

software used to operate the spectrometer can also perform FFTs, we again chose I2S to maintain consistency. A developmental version of I2S was used, which was adapted to also allow FFT processing on EM27/SUN interferograms. I2S splits a raw forward/backward ifg into two different double-sided ifgs which are then FFTed to yield two spectra. I2S also corrects source brightness fluctuations (Keppel-Aleks et al., 2007).

### 3.2 EM27/SUN GGG and I2S Suite (EGI)

To make GFIT retrievals simpler for new EM27/SUN users, an add-in software suite (EGI) was developed at Caltech to create correctly formatted input files. This suite is open-source and can be obtained through correspondence to the email address listed. EGI can be run using MATLAB or Python. EGI runs in UNIX, Mac OS, and Linux environments and runs I2S and GFIT on multiple

processors. EGI centralizes settings for paths to read and write files, it coordinates separately acquired ground weather station and GPS data with EM27/SUN ifgs, and it optimizes processing order. It also





provides some ancillary calculations such as a spectral signal-to-noise ratio (SNR) calculation. EGI provides a simple way to turn on and off saving of ancillary retrieval files (i.e. spectral fits and averaging kernels). EGI is automated, reducing the learning time as well as the amount of user time needed to retrieve DMFs. After an initial setup, EGI will run from ifgs to retrieved $X_{gas}$ with two

commands. On a computer with 1400 MHz processors the code takes ~30 s per CPU to process each interferogram from the EM27/SUN extended InGaAs detector.

## 4 Instrument Characterizations and Performance

## 4.1 Instrument Line Shape

Knowledge of the instrument line shape (ILS), or the observed shape of a spectral line from a

monochromatic input, is crucial in assessing instrument performance. Two parameters are used to characterize the ILS in relation to an ideal instrument, namely the modulation efficiency (ME) and phase error (PE). ME and PE both describe the interferogram and vary with OPD (Hase et al., 1999; Frey et al., 2015). PE is shift in the interferogram from the sampling comb, with an ideal value of 0 radians, and causes asymmetry in spectral lines. ME is a measure of the normalized observed

interferogram signal compared with that of an nominal instrument with an ideal value of 1 (unitless) (Hase 2012). At maximum OPD (MOPD), an ME<1 causes a measured spectral line to be broader than an ideal line and indicates angular misalignment, while an ME>1 at MOPD causes a narrower measured line than ideal and indicates shear misalignment. The ILS can be calculated by analyzing absorption lines measured through a low-pressure gas cell, and varies with OPD (Hase et al., 1999). Here, we use

only single ME and PE values at the MOPD (Frey et al., 2015) to describe the ILS. We characterized the ILS for the EM27/SUN instruments using the method described elsewhere (Frey et al., 2015; Klappenbach et al., 2015). This method is able to characterize ME to within 0.15% using the LINEFIT algorithm (Hase et al., 1999), with supplemental MATLAB scripts for automation purposes (Chen et al., 2016). ILS can affect retrieved column values. We note that the ME at MOPD of the cn and ha

instruments in Table 2 are significantly lower than those reported by KIT on campus of ~0.997 (Frey et al., 2015), and post-campaign of ~0.996 (Klappenbach et al., 2015).





For this study, the ILS is used to help explain biases, to demonstrate the stability of the instruments, and gives insight into how well the EM27/SUN instruments are aligned and their optical aberrations. Though GGG2014 retrievals do not account for non-ideal ILS, future versions of GGG will. For the current study, we assume that ILS impacts using PROFFIT will be similar to impacts using GFIT. This assumption will need to be tested when GFIT also can account for a non-ideal ILS. Because future GFIT retrievals will be revised using historical ILS measurements, there remains a need to monitor the ILS both for future retrievals and as an indicator if realignment is necessary.

## 4.2 Frequency shifts

EM27/SUN units contain a standard HeNe 633 nm (15798 cm$^{-1}$) metrology laser to sample the IR signal accurately as a function of the OPD. The laser is not frequency stabilized (Gisi et al., 2012). This causes apparent spectral frequency to change with temperature as is shown in Fig. 2. Frequency shifts are affected by changes in the input laser wavenumber, laser alignment, and IR beam alignment. The input laser wavenumber will affect the spacing between spectral points. Since the frequency shift is furthest from zero for the Caltech EM27/SUN (on order of -100 ppm, in red Fig. 2), the spectral spacing is empirically corrected in the EGI suite based on the $CO_2$ 6220 cm$^{-1}$ window frequency shifts. This made little difference for the primary gases of interest affecting $X_{CO2}$ by 0.015% and $X_{CH4}$ by -0.005%, though it did affect $X_{H2O}$ by 4%.

## 4.3 Ghosts

Ghosts are artificial spectral features linked to aliasing of true spectral lines that arise in FTS spectra (Learner et al., 1996). The InGaAs detectors are optically sensitive at wavenumbers greater than half the HeNe metrology laser frequency (7899 cm$^{-1}$). To fulfill the Nyquist criterion and prevent aliasing, the IR interferogram is sampled twice each laser interferogram cycle, on the rising and falling edge. However, if the laser sampling is asymmetric—for example from a faulty electronics board—aliasing can still occur, folded across the half laser frequency. Because the asymmetry is typically small, the aliased signal, or ghost spectrum, is small compared with the true spectrum (Dohe et al., 2013; Wunch et al., 2015).



In EM27/SUN instruments the laser sampling error (LSE) can be minimized as data are collected by employing the interpolated sampling option provided by Bruker[TM] which only uses the rising edge of the laser interferogram and assumes constant velocity in between the rising edges to interpolate the sampling. We use a narrow-band pass filter (3 dB band-width 5820–6150 cm$^{-1}$) in the Caltech

EM27/SUN to test for LSE ghosts at 9800 cm$^{-1}$. The ghost to parent ratio (GPR) is $1.73 \times 10^{-4}$ at a 10 kHz acquisition rate without the interpolated sampling activated. This ghost is eliminated with the interpolated sampling turned on. In actual solar tests, turning the interpolated sampling on and off had no noticeable effect on the DMF retrievals for the Caltech EM27/SUN, however this may not hold true for all instruments. The LSE ghost also disappeared at an acquisition frequency of 20 kHz, and returned

at higher acquisition frequencies. We opted for the recommended 10 kHz acquisition rate with the interpolated sampling on for all EM27/SUNs in this analysis because other instruments may be more significantly affected by LSE ghosts. There remains a double-frequency ghost at ~11900 cm$^{-1}$ from radiation passing through the interferometer twice that is much larger than the LSE ghost, but is not in a region that will affect retrievals.

**4.4 Mirror Degradation and Detector Linearity**

Solar tracking mirrors provided with the EM27/SUN instruments are protected gold coated. Gold is used because of its excellent reflectance in the near-IR and low reflectance in the visible region (Bennett and Ashley, 1965), which allows high signal while reducing excess heating of the field stop and other optics. Through extended tests, we noted the first two mirrors (gold on plated aluminum, with

a coating) degrade over time, with an e$^{-1}$ (e-folding) degradation time of ~90 days as is shown in Fig. 3. Cleaning helped restore some signal, but never to the original values. The mirror change may not have restored full signal because the rest of the optics were not cleaned at the time of the mirror change. Below the blue 150 arbitrary unit (AU) line in Fig. 3 the fitted $O_2$ RMS as a percentage of the continuum level dropped 26 times faster with signal intensity than above it. The instrument did come

with an extra set of mirrors, but because mirrors are consumable parts it adds recurring cost and effort to maintain these instruments long-term. After a year of use, the third mirror (gold coated glass) still remains completely intact. Feist et al. (2015) had success using steel mirrors under the very harsh



conditions at the Ascension Island TCCON site, though at a cost of 35% reflectivity per mirror. The JPL TCCON sites near Caltech noted no degradation on the external gold mirrors over more than a year of measurements. Mirror degradation has likely not been a widely reported problem for most of the EM27/SUN community perhaps because these instruments typically are stored indoors and only used

for a few days for campaigns (for example, Frey et al., 2015). However, this problem may affect mirrors on other EM27/SUN instruments when mirrors are exposed outside for extended periods of time.

With signal loss, we would anticipate that gas measurements would become noisier but remain unbiased. However, with time the Caltech EM27/SUN $X_{CO2}$ and $X_{CH4}$ DMFs decreased relative to the TCCON DMFs as mirror reflectance decreased, and $X_{CO2}$ and $X_{CH4}$ increased when the mirrors were

10 replaced. The TCCON 125HR InGaAs detectors are already known to be sufficiently linear that no correction is required (Wunch et al., 2011b). We also performed a simple test repeatedly adding mesh screens in front of the entrance window to filter some of the light. In these tests $X_{CO2}$ and $X_{CH4}$ changed on order of 3 ppm and 0.01 ppm respectively when using the extended-InGaAs detector in the presence of filters transmitting ~25% of the light. Figure 4 shows results from this test on $X_{CO2}$; results from $X_{CH4}$

are similar. This provides strong evidence that the extended-InGaAs detector is non-linear. We repeated the test using the standard InGaAs detector and changes in $X_{CH4}$ and $X_{CO2}$ biases were on order of 10× smaller and could be attributed to scattering off the mesh screen placed in front of the entrance window. Figure 5 shows the difference between the EM27/SUN and TCCON $X_{CO2}$ and $X_{CH4}$ as the total signal changed. After the mirrors were changed the relative difference actually went up with some signal loss

before decreasing again, for reasons we do not understand.

Detector non-linearity in FTS instruments can be corrected in the ifgs post-acquisition in 2 ways. The first option deals with artifacts around the ZPD and is already included in I2S (Keppel-Aleks et al., 2007). When the ifg is smoothed, a non-linear detector exhibits a dip around the ZPD which can be used to diagnose and reduce detector non-linearity effects. EM27/SUN measurements are too noisy to

25 properly characterize or detect this dip and so this correction is insufficient. The other option is to compare detector response with radiance from a controlled external light source, such as a blackbody, with very accurate radiation flux measurements (on order of 0.01%) (Thompson and Chen, 1994). By characterizing the response to the true flux as it is varied, the detector can be characterized and ifgs can



be appropriately scaled and corrected. However, this requires extremely controlled precise measurements as all non-linearity is likely less than 1%, so measurements must be more precise than 1%.

An option to prevent non-linearity from interfering with measurements is to only use the detector over its linear range by sufficiently attenuating the incoming sunlight. However, the SNR is already low so we opted against this method. Ultimately, we purchased the non-extended InGaAs detector at the loss of CO, and $N_2O$ for future measurements for the Caltech instrument. For the historical field measurements we use a bias correction to match the TCCON for the nearest comparison days. The non-linearity has nearly an equal effect for short times, but has a larger variation on multi-monthly scales as the mirrors degrade. In future measurements we recommend against using these extended InGaAs detectors. Addition of band-pass filters or use of different detectors will be necessary to provide high quality measurements of CO, $CO_2$, and $CH_4$.

The data shown in Fig. 5 were divided into bins based on the signal intensity and were separated before and after the mirror change. Within each bin the relationship was treated as approximately linear. Fits using less than 10 points or with correlation coefficients less than 0.1 were discarded. The change with half signal was calculated. The analysis was repeated for 10 bins and again for 20 bins. The weighted mean change in $X_{CO2}$ for halving the signal is -1.43 ppm in agreement with the mesh tests or

$$\Delta X_{CO2}(\text{ppm}^{-1}) = 2.06 \ln(S/S_0) \tag{2}$$

where S and $S_0$ are the final and initial signals respectively. This relationship holds for S and $S_0$ in the middle 80%. For a similar methane analysis the mean change for half signal is -7.25 ppb or

$$\Delta X_{CH4}(\text{ppb}^{-1}) = 10.5 \ln(S/S_0) \tag{3}$$

## 5 Comparisons with $X_{gas}$

GGG2014 includes an airmass dependent correction factor derived for TCCON $X_{gas}$ measurements. The airmass correction factor for each gas is calculated using data obtained at a variety of relatively clean sites as described by Wunch et al. (2011b). We expect that the airmass dependence, which is due primarily to spectroscopic uncertainties, should be common for the same type of measurement. Parker et al. (2016) showed the EM27/SUN factors are similar compared to the TCCON for $X_{CO2}$ at 3 clean



sites in the U.S. The $X_{CH4}$ β factor was different (-0.0077 EM27/SUN, 0.0053 TCCON) but when applied here it worsened the $R^2$ and standard deviation of the comparisons. This could be because the airmass dependence of $X_{CH4}$ may not be solely from spectroscopic issues. Hence, we used the same airmass dependent correction factors as the TCCON.

To compare measurements between the TCCON and the EM27/SUN instruments, data were first averaged into 10 minute bins to reduce the variance of binned differences (Chen et al. 2016). The median of the $X_{CO2}$ differences between sequential time bins is smallest (around 0.26 ppm) for 10 minute bins over the entire ~11 month time period. Less averaging is more affected by noise, and more averaging starts to include instrument drift and true atmospheric variations. Averages were weighted

using retrieval errors $\hat{x}_{err}$ as in Eq. (4):

$$\hat{\hat{x}} = \frac{\sum_i \hat{x}_i \hat{x}_{i,err}^{-2}}{\sum_i \hat{x}_{i,err}^{-2}} \tag{4}$$

where $\hat{x}_i$ is the retrieved value from the $i$th measurement in a bin, and $\hat{\hat{x}}$ is the bin average.

## 5.1 Averaging Kernels

When comparing retrieved $X_{gas}$ measurements (also denoted $\hat{c}$) from different remote sensing instruments, differences in their averaging kernels (AKs or $\mathbf{a_i}$ where i represents an instrument indicator

number) and a priori profiles must be taken into account, using for example, the methods described by Rodgers and Connor (2003). Wunch et al. (2011a) compared GOSAT and TCCON total column DMFs using this method. Because GFIT scales a priori profiles rather than retrieving the full profile these AKs are vectors (i.e., column averaging kernels) rather than matrices.

Averaging kernels depend on several factors including how strong the lines are in the retrieval

windows, and viewing geometry (e.g. solar zenith angle (SZA) for solar-viewing instruments). Because the TCCON 125HR instruments and EM27/SUN instruments have different spectral resolutions, the apparent absorption strengths are different and so are the averaging kernels. Averaging kernels for a gas differ for each microwindow. We combined AKs of a given gas from different microwindows using an unweighted average. Averaging kernels for the Caltech EM27/SUN for the GFIT retrieval windows are

shown in Fig. 6. Averaging kernels from the other EM27/SUN instruments are similar. TCCON



averaging kernels have been discussed by Wunch et al. (2011b) and are shown on the bottom row in Fig. 6. As a numerical example, for $X_{CO2}$ measured at 50° SZA and 900 hPa using GFIT, the AK is 1.10 for EM27/SUN instruments and 0.93 for TCCON instruments. This means EM27/SUN instruments are slightly more sensitive to a change in $CO_2$ near the surface relative to TCCON instruments. More importantly, they have the opposite sensitivity to an error in the a priori volume mixing ratio (VMR) profile at 900 hPa.

In our particular case, reducing the smoothing error using Eq. A13 from Wunch (2011a) and using the a priori as the comparison ensemble changes little as the effect of the differences in averaging kernels from the top of the atmosphere tends to cancel out the effect of differences at the bottom. TCCON and EM27/SUN a priori profiles were the same in this comparison. However, we need to consider that the a priori profiles used in the retrieval are not representative of a highly polluted place, such as Pasadena, which is located in the same air basin as Los Angeles. Because differences in column measurements compared to background or a priori profiles occur primarily because of differences at the surface we can adjust retrievals for one instrument taking into account this knowledge using:

$$\hat{c}_1 = \frac{a_{1,s}}{a_{2,s}}[\hat{c}_2 - c_a] + c_a \tag{5}$$

Definitions of the terms in, as well as a discussion of assumptions needed to obtain Eq. 5 are in Appendix A. We applied Eq. 5 to the $X_{CO2}$ and $X_{CH4}$ retrievals.

In summary, to compare biases between two instruments, we account for diurnal dependences, then average data into comparable time bins, and take into account our prior knowledge of the atmospheric profile and differences in averaging kernels.

## 5.2 Full comparisons of $X_{gas}$ from extended-InGaAs detector with a TCCON site

Gisi et al. (2012) noted that measurements taken within the first 30 minutes of moving the instrument to the roof and turning it on needed to be filtered out because of high scatter while waiting for the instrument to operate stably. We did not observe a similar requirement for our data. This could be because our instruments were not subjected to such fast temperature changes. It could also be because the laser frequency shift, which changes with temperature, does not seem to significantly impact our retrievals.





The full time series (186 days) of the difference between the Caltech EM27/SUN and TCCON measurements is shown in Fig. 7. From this figure we see $X_{CO2}$ and $X_{CH4}$ are the gases most affected by the mirror change in October 2014 (by about 3 ppm and 12 ppb respectively). For all gases, scatter of retrieved $X_{gas}$ increases as signal decreases. Figure 8 shows the retrieved $X_{CO2}$ and $X_{CH4}$ from all 4

EM27/SUN instruments for 9–12 days in January 2015 plotted against those from TCCON. We report biases as scaling factors to approximate to the TCCON, or scaling factors compared to 1. Biases were calculated using a linear least squares fit forced through the origin. A summary of the biases for all gases as compared to the TCCON is provided in Table 3.

### 5.3 $X_{CO2}$

We note a smaller bias in $X_{CO2}$ with respect to the TCCON (+0.03%, see Table 3) compared to previous EM27/SUN studies (Gisi et al., 2012; Frey et al., 2015; Klappenbach et al., 2015). These previous studies retrieved $X_{gas}$ from EM27/SUN spectra using PROFFIT. When compared with the TCCON $X_{CO2}$ retrievals, Gisi et al. (2012) noted a +0.12% bias, Frey et al. (2015) noted a +0.49% bias, and Klappenbach et al. (2015) noted a +0.43% bias. Reasons for these differences could be from 1)

spectroscopy differences between PROFFIT and GGG2014 used for EM27/SUN $X_{gas}$ retrievals, 2) because Gisi et al. (2012) used an earlier version of GFIT for TCCON retrievals and 3) because Frey et al. (2015) and Klappenbach et al. (2015) applied empirical corrections before comparing with the TCCON. In this section, we investigate two possible causes of bias: spectral resolution and instrument line shape.

Following Gisi et al. (2012), we attempted to determine whether the cause of the bias is due to the difference in spectral resolutions between the EM27/SUN and TCCON instruments. Petri et al. (2012) also considered resolution bias in their study using a 0.11 cm$^{-1}$ resolution instrument and an older version of GFIT. They did not report a bias in $X_{CO2}$ retrievals, but noted that $X_{CO2}$ decreased by ~0.12% as interferograms were truncated to obtain spectra with resolutions of 0.02 cm$^{-1}$ to 0.5 cm$^{-1}$. Most of the

change occurred as the resolution changed from 0.1 cm$^{-1}$ to 0.5 cm$^{-1}$ (see Fig. 11 therein). In contrast, Gisi et al. (2012) noted a 0.13% increase in $X_{CO2}$ as the resolution changed from 0.02 cm$^{-1}$ to 0.5 cm$^{-1}$ in PROFFIT. Here we find a small 0.02% ± 0.13% (1σ) increase in $X_{CO2}$ when the resolution is decreased



from 0.02 cm$^{-1}$ to 0.47 cm$^{-1}$ in GFIT, though part of this change would be offset by considering the differences in averaging kernels.

Previous studies noted an increase in $X_{CO2}$ of 0.15% for a 1% increase in modulation efficiency at max OPD (Gisi et al., 2012; Frey et al., 2015). Using PROFFIT we performed a similar test for spectra taken

under various conditions at various times of day and obtained a similar result of a 0.10% ± 0.02% (1σ) increase in $X_{CO2}$ for a 1% increase in ME at the MOPD. For this study we assume impacts of the ILS on retrievals will be similar in GFIT and PROFFIT. Though we report a single value, there is an airmass dependence of ~0.05% increase in EM27/SUN PROFFIT retrievals for a 1% increase in ME and airmass change of 1.

For instruments using the standard InGaAs detectors the $X_{CO2}$ 10-minute running 1σ precision is 0.075% [0.034% to 0.18%, 95% CI]. The wide confidence interval (CI) is from a combination of atmospheric variability being aliased in as well as different SNRs among instruments. The spectral SNRs for measurements using this detector were in the range 1000–5000 and their precision for $X_{CO2}$ retrievals was only weakly correlated with $1/\sqrt{SNR}$. Chen et al. (2016) found the 1σ $X_{CO2}$ precision

among 10-minute binned EM27/SUN$_a$-EM27/SUN$_b$ differences is 0.01%. These data were acquired in a way that about 67 spectra were acquired every 10 minutes, and because 2 instruments were used the single sounding precision is ~0.01% × $\sqrt{67/2}$ ≈ 0.058%, which falls in our measured running 1σ precision range. Comparing to the TCCON, Gisi et al. (2012) reported that the 1σ daily precision is 0.08%. The extended-InGaAs detector naturally has a lower spectral SNR, in the range 100–1000, with

a median of 400 over the full time series. Most of the variation in the SNR is due to loss of mirror reflectivity, but even with non-degraded gold mirrors is ~5× lower because of the different detector. The median running 1σ precision over the full time series is 0.26% for the $X_{CO2}$ product from the extended InGaAs detector. Because the SNR changed with time due to loss of mirror reflectivity, so did the precision. The correlation between $1/\sqrt{SNR}$ and running 1σ $X_{CO2}$ precision was strong ($R^2$=0.75) for

retrievals from this detector and followed

$$\sigma_{XCO2} = 0.17 + \frac{8.4}{\sqrt{SNR - 57}} \tag{6}$$





An additional study we have not performed that could help in reducing bias would be to omit all or part of a $CO_2$ window with strong water lines. Because of the low resolution of these spectrometers (see inset Fig. 1), water lines and $CO_2$ lines are often overlapping. This can lead to inaccurate retrievals despite a good overall fit because $H_2O$ and $CO_2$ can both be wrong, but in compensating ways. Reducing the size of a window would reduce precision but would decrease water and temperature sensitivity. This adjustment could also be performed for $CH_4$ which is retrieved over three windows in GFIT.

## 5.4 $X_{CH4}$

The EM27/SUN $X_{CH4}$ retrievals are 0.75% higher than those of TCCON (see Table 3). In previous work, high biases of 0.47% for a 0.11 cm$^{-1}$ instrument (Petri et al., 2012), and 0.49% (Frey et al., 2015) and 1.87% (Klappenbach et al., 2015) for EM27/SUNs were noted. Petri et al. (2012) attributed most (0.26%) of their bias to differences in resolution and noted for a single day that the bias increased as resolution decreased. In our simulations we find a 0.20% ± 0.15% (1σ) increase in $X_{CH4}$ when the resolution is reduced from 0.02 cm$^{-1}$ to 0.47 cm$^{-1}$. Using PROFFIT the impact of a 1% decrease in ME is a 0.15% ± 0.01% (1σ) increase in $X_{CH4}$. Again, although we report a single value there is an airmass dependence of about a 0.12% decrease in $X_{CH4}$ using PROFFIT retrievals for an airmass change of 1, and a 1% decrease in ME. Resolution and ME combined account for only half of the observed methane bias. Petri et al. (2012) suggested improper dry air mixing ratio and pT profiles, or spectroscopy as sources of error. Improper surface pressure, error in the calculated Observer-Sun Doppler Stretch (OSDS) due to pointing errors coupled with solar rotation, or error in the assumed field of view (FOV) may also contribute to the bias (see §6).

Chen et al. (2016) found the 1σ $X_{CH4}$ precision among 10-minute binned EM27/SUN$_a$-EM27/SUN$_b$ differences is 0.01%, which is equivalent to a single sounding 1σ precision of ~0.058%. Using the same method as for $X_{CO2}$, the $X_{CH4}$ running 1σ precision from instruments using the standard InGaAs detectors is 0.057% [0.037% to 0.25%, 95% CI], in agreement with Chen et al. (2016). The median running 1σ precision for $X_{CH4}$ from instruments using the extended InGaAs detector is 0.33%. $X_{CH4}$ precision from the extended InGaAs measurements is also correlated with $1/\sqrt{SNR}$.



## 5.5 $X_{CO}$ & $X_{N2O}$

Along with $X_{N2O}$, $X_{CO}$ was measured for the first time using an EM27/SUN spectrometer in this study. Column CO measurements are desirable because CO is a tracer of combustion. These measurements were made possible because the extended detector is sensitive to the region 4200–4800 cm$^{-1}$ which contains useful windows where $N_2O$ and CO molecules absorb IR radiation. Both the $X_{CO}$ and $X_{N2O}$ retrievals are highly sensitive to changes in the modeled temperature profile. The non-linearity of the detector had a less pronounced effect on $X_{CO}$ and $X_{N2O}$ retrievals than it had on $X_{CO2}$ and $X_{CH4}$ retrievals. $X_{CO}$ and $X_{N2O}$ also have poorer precision than $X_{CO2}$ and $X_{CH4}$ so any non-linearity effect could be lost in the noise. The 4200–4800 cm$^{-1}$ spectral region is also affected differently than the 5000–7000 cm$^{-1}$ region where column $CH_4$ and $CO_2$ are retrieved from and may also explain in part why there is no noticeable change in $X_{CO}$ and $X_{N2O}$ with signal. For $X_{CO}$ the median $1\sigma$ precision is 3.7%. In our simulations reducing the spectral resolution from the TCCON (0.02 cm$^{-1}$) to the EM27/SUN (~0.5 cm$^{-1}$) $X_{CO}$ increases $1.1\% \pm 0.9\%$ ($1\sigma$) in low resolution spectra.

In general, as is seen in Fig. 7, $X_{N2O}$ retrievals were highly scattered and had a large offset from TCCON. In our simulations reducing the resolution from TCCON (0.02 cm$^{-1}$) to EM27/SUN (~0.5 cm$^{-1}$) decreased $X_{N2O}$ by $2.2\% \pm 0.6\%$ ($1\sigma$). Retrievals from the 4430 cm$^{-1}$ window were low (~6%) while the 4719 cm$^{-1}$ and 4395 cm$^{-1}$ regions were biased slightly high (~1%). The retrievals from the 4719 cm$^{-1}$ region additionally had some long term trends for reasons we do not understand. For $X_{N2O}$ the median $1\sigma$ precision is 1.9%.

## 5.6 $X_{H2O}$

Because of the significantly lower spectral resolution of the EM27/SUN spectrometers, the spectral band widths for the $H_2O$ retrievals were increased as compared to the standard TCCON approach (Wunch et al., 2010). For lower resolution spectra, the $H_2O$ lines appear much broader and the observed transmittance is much lower at the edges of standard TCCON spectral window. Thus, the spectral ranges of the low-resolution windows were expanded. Some of the standard TCCON windows used to retrieve $H_2O$ had too few spectral points from the low-resolution instrument for good fits and were omitted. When expanding the windows, we ensured that no lines were admitted that made the effective



ground-state energy E'' greater than ~400 cm$^{-1}$. This reduces the temperature sensitivity to the modeled temperature profiles. As with the TCCON windows, we tried to keep a wide range of $H_2O$ line strengths to accommodate large seasonal and site-to-site variations of the $H_2O$ column. Windows were kept as wide as possible without encountering large spectral fitting residuals.

For $X_{H2O}$, we find a median $1\sigma$ precision of 1.9% from the instrument using the extended InGaAs detector. For instruments using the standard-InGaAs detectors, the $X_{H2O}$ $1\sigma$ precision is 0.81% [0.36% to 2.12%, 95% CI].

## 6 Sensitivity tests on retrievals

As with TCCON, EM27/SUN retrievals require modeled atmospheric pressure, temperature, altitude
(pTz) and water profiles (Wunch et al., 2015). Here atmospheric profiles are generated from the NCEP/NCAR 2.5° reanalysis product (Kalnay et al., 1996) by interpolating to the correct location at local noon of the desired day. These profiles also include the tropopause height which is used to vertically shift a priori profiles, as tropopause height can significantly affect column DMFs such as $X_{CH4}$ and $X_{HF}$ (Saad et al., 2014). Selecting a profile for an incorrect location or day could lead to errors.
We ran test retrievals for the July 2014 period with incorrect profile information derived separately at latitudes north (1, 2, and 5 degrees) and longitudes west (1, 2, and 5 degrees) of our observation site, and well as from profiles derived 1, 5, 10 and 100 days prior to the measurement dates. In general, the profiles generated from a more distant location in space and time caused larger retrieval errors. For $X_{CH4}$, and $X_{CO}$ the main variability from the standard retrievals was in daily offsets (standard deviation
of daily medians $\sigma(Md_{daily})$) which had values of 3 ppb and 4 ppb respectively for the 100 day prior model. The medians of daily standard deviations $Md(\sigma_{daily})$ were 0.5 ppb for both $X_{CH4}$ and $X_{CO}$ for the 100 day prior model. $X_{N2O}$ and $X_{H2O}$ also had more errors from $\sigma(Md_{daily})$, except for profiles within 2 degrees, which more strongly affected diurnal variability $Md(\sigma_{daily})$. For these 2 species the 100 day prior model $\sigma(Md_{daily})$ were 2 ppb and 50 ppm and $Md(\sigma_{daily})$ were 1 ppb and 20 ppm respectively.
These values are shown for $X_{CO2}$ in Fig. 9 for all tested models. The 100 day prior model had



$\sigma(\text{Md}_{\text{daily}}) = 0.16$ ppm and $\text{Md}(\sigma_{\text{daily}}) = 0.4$ ppm as well as a 1.2 ppm bias when using these models for $X_{CO2}$.

Various user, instrumental, and measurement errors can reduce the accuracy and precision of retrievals. GFIT uses retrieved $O_2$ column amount with the average DMF of $O_2$ (0.2095) to calculate the dry

pressure column of air. However, to calculate the $O_2$ absorption coefficients, GFIT takes into account the surface pressure, which can lead to measurement inaccuracies if the wrong surface pressure is used. Wunch et al. (2011b) reported a 0.04% $X_{CO2}$ bias for a +1 hPa surface pressure offset in the TCCON. Similarly, we find a 0.032% $X_{CO2}$ bias per +1 hPa surface pressure offset with a 0.004% $\sigma$ variation on average throughout a day. Because the pressure offset affects $O_2$ retrievals, the other species are also

affected (Table 4). $X_{CO}$ may be particularly affected by a pressure bias because such a large fraction of the column CO is near the surface.

Using the same July 2014 dataset used to test the sensitivity of the retrievals to error in the pTz profile and surface pressure, we further estimated the sensitivity to error in the temperature in the lower atmosphere (surface–700 hPa). GFIT uses a single temperature profile per day that represents the

local-noon temperatures and the surface temperature is extracted from that profile. Such temperature error can arise in particular at the beginning and end of the day when the temperature is typically cooler than at noon. Here we derived the sensitivity of the retrievals to a +10 K error in the lower atmosphere (Table 4). $X_{CO}$ has a significantly larger bias than the other species, likely because water absorption lines are the strongest spectral features in the CO retrieval window and water absorption lines are highly

sensitive to changes in temperature. Water lines are also much stronger than $N_2O$ lines in the $N_2O$ windows. These tests suggest offsets under 1 hPa and 1 K would cause small (~0.1 ppm) biases on $X_{CO2}$, but a 4 K difference in near surface (ground–700 hPa) temperature could cause ~0.4 ppm bias in $X_{CO2}$ which is larger than our reported $1\sigma$ precision. For other studies using multiple spectrometers and multiple meteorological measurements for $X_{gas}$ retrievals, we recommend cross-comparing

meteorological measurements to eliminate bias—preferably to a standard.

Finally, we perform a sensitivity study following the methodology of Wunch et al. (2015). The magnitudes of the applied perturbations are in Table 5. The results of this uncertainty budget study are presented for a day for $X_{CO2}$ and $X_{CH4}$ in Fig. 10. We do not include a sum in quadrature because we do



not have an exhaustive list of sources of uncertainty. Some of these errors may partially account for the unexplained long-term drifts we noted compared to TCCON. For example, surface pressure and calculated Observer-Sun Doppler Stretch (OSDS) were correlated with EM27/SUN to TCCON $X_{CO2}$ differences in the long-term measurement. However, there was no apparent trend in the spectral

residuals from fitting solar lines as the OSDS changed so this correlation may not indicate cause. This uncertainty budget indicates that the low resolution instruments are especially sensitive to biases in a priori pressures and a priori volume mixing ratio (VMR) profiles.

## 7 Conclusions

Despite the challenge associated with the extended InGaAs detector and mirror degradation, the

EM27/SUN instruments perform well on short time scales with 1σ precisions of 0.075% for $X_{CO2}$ and 0.057% for $X_{CH4}$ retrieved from measurements using the standard InGaAs detectors. These instruments perform well in terms of mobility and stability, maintaining alignment despite frequent movement and jostling—an ideal characteristic of mobile FTS instruments. Measurements from the standard detector are precise enough to be used for campaigns of up to a few months and to provide useful supplementary

$X_{gas}$ measurements to established networks like TCCON. However, we recommend regular—6 months to a year depending on use—comparison with established measurements (e.g. a TCCON site) to account for long-term drift. Simultaneous use of several EM27/SUN instruments can also help characterize drift. We also recommend regular—about monthly depending on use—ILS characterization. Our experience also suggests that the extended InGaAs detector is incompatible with precise $X_{CO2}$ and $X_{CH4}$ retrievals.

In general, we recommend all new ground-based, solar-viewing, remote sensing FTS instruments to undergo some or all tests listed in Table 6 to evaluate their performance. We also recommend comparisons of retrieval outputs to those of existing instrumentation (e.g. TCCON or NDACC-IRWG). These tests assume that one of the 3 widely used and accepted retrieval algorithms (GFIT, PROFFIT, and SFIT) which are known to provide accurate spectral fitting is used. New retrieval algorithms should

be subjected to additional comparisons with currently accepted algorithms. Some of the results of these tests will be similar across all instruments of a given type, and so do not need to be repeated if they have been performed on another instrument elsewhere.





## Appendix A

### Assumptions and limitations in the AK correction

To derive Eq. 5, we begin with Eq. 22 in Rodgers and Connor (2003).

$$\hat{c}_i = c_a + \sum_k h_k a_{i,k} \cdot (x_{t,k} - x_{a,k}) + \epsilon_i \tag{A1}$$

To include the pressure-weighting function $\boldsymbol{h}$ (Connor et al. 2008) we have used summation notation.
The "hat" represents a retrieved value, and $c$ represents a column (scalar) value, and $\epsilon$ is the error.
Subscript $i$ is for a particular instrument, subscript $a$ represents the a priori, subscript $k$ is for a
particular atmospheric layer, and subscript $t$ represents the true atmosphere. The vectors $\boldsymbol{a}$ and $\boldsymbol{x}$
represent the column averaging kernel and atmospheric VMR profile respectively. This equation is
derived from Eq. 1 in Rodgers and Connor (2003) using a Taylor series expansion about the a priori,
and assuming linearity about it.

To compare retrievals from remote sounding instruments, a comparison profile (also called the
comparison ensemble mean, denoted $\boldsymbol{x}_c$) is used. Here, we have used the daily a priori profiles, which
were the same for all instruments, as the comparison profiles. We note, however, that the comparison
profiles should describe the real atmosphere as far as possible (Rodgers 2000). Though the a priori has a
draw down in $CO_2$ from the biosphere near the surface, the real atmosphere in Pasadena is polluted near
the surface. Thus this choice of comparison profiles is not ideal in our situation.

If we ignore retrieval error Eq. A1, and further assume that $\boldsymbol{x}_t = \boldsymbol{x}_a$ except at the surface it can be
rewritten as:

$$\frac{1}{a_{i,s}} (\hat{c}_i - c_a) = h_s (x_{t,s} - x_{a,s}) \tag{A2}$$

Where the subscript $s$ represents a surface value. If we are comparing measurements from two different
instruments, $i = 1$ and $i = 2$, in the same location, $x_{t,s}$ and $h_s$ are the same. Because the a priori
profiles are also the same

$$\frac{1}{a_{1,s}} (\hat{c}_1 - c_a) = \frac{1}{a_{2,s}} (\hat{c}_2 - c_a) \tag{A3}$$

Which can be rewritten as



$$\hat{c}_1 = \frac{a_{1,s}}{a_{2,s}}(\hat{c}_2 - c_a) + c_a \tag{A4}$$

Even in the absence of error, retrievals from instruments with different averaging kernels will still differ.

We adjust the EM27/SUN $X_{CO2}$ and $X_{CH4}$ retrievals using Eq. A4 before comparison with the TCCON, which adjusts $X_{CO2}$ by up to ~1.2 ppm, and $X_{CH4}$ by up to ~8 ppb. Future work could improve on this methodology using a better comparison ensemble or more representative a priori profiles for retrievals from measurements in Pasadena. This correction is not applied to $X_{H2O}$ because the AKs vary more among spectra because of larger variations in absorption strengths. It is also not applied to $X_{CO}$ and $X_{N2O}$ because using $\boldsymbol{x}_t = \boldsymbol{x}_a$ is too poor of an assumption and makes the comparison worse between the TCCON and EM27/SUN retrievals in terms of $R^2$.

## Acknowledgements

We thank Frank Hase and Michael Gisi for helpful discussions on ghost reduction, detector non-linearity, and ILS measurements. We further thank Michael Gisi and Bruker Optics$^{TM}$ for loaning us a standard InGaAs detector for testing and for instructions on realigning the EM27/SUN. We thank Dietrich Feist for discussions on mirror degradation. We also thank Nicholas Jones, David Giffith, Frank Hase, and Sabrina Arnold for sharing their experience with mirror degradation. This work is supported in part by the W. M. Keck Institute for Space Studies. Jacob Hedelius was also partially supported from a Caltech Chemistry and Chemical Engineering Division Fellowship funded by the Dow Chemical Graduate Fellowship and thanks them. The authors gratefully acknowledge funding from the NASA Carbon Cycle Science program, and the Jet Propulsion Laboratory. MKD acknowledges funding from the NASA-CMS program for field observations and from the LANL-LDRD for the acquisition of the LANL EM27/SUN.

The authors declare no conflict of interest.



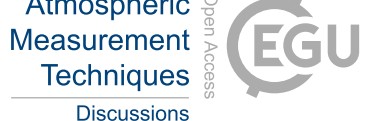

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





Table 1. Pre-averaging and apodization effects on EM27/SUN retrievals.

| % Error | $X_{CO2}$ | | $X_{CH4}$ | | $X_{H2O}$ | | $X_{CO}$ | | $X_{N2O}$ | |
|---|---|---|---|---|---|---|---|---|---|---|
| | Md. | σ | Md. | σ | Md. | σ | Md. | σ | Md. | σ |
| 5 fwd/bwd pre-avgd.[a,b] | <0.01 | 0.01 | -0.02 | 0.01 | 0.36 | 0.13 | <0.01 | 0.15 | 0.30 | 0.12 |
| NB med. apodz.[a,c] | 0.29 | 0.09 | -0.07 | 0.10 | 0.35 | 0.23 | -1.01 | 0.58 | -1.36 | 0.55 |

Measurement compared over 1 July–10 July 2014. Md=Median. NB=medium Norton-Beer apodization. [a]As compared to retrievals from 1 fwd/bwd averaged non-apodized measurement averaged over same time post-retrieval. [b]Same apodization as standard. [c]Same pre-averaging as standard.



Table 2. ILS of EM27/SUN instruments

| Instrument Num – ID | Jan 9 2015 ME PE (mrad) | Jan 28 2015 ME PE (mrad) |
|---|---|---|
| Caltech | 0.985 | 0.975[a] |
| (42 – cn) | 4.87 | 3.44 |
| LANL | 0.999 | |
| (34 – pl) | -1.35 | |
| Harvard 1 | 0.977[b] | |
| (45 – ha) | -2.17 | |
| Harvard 2 | 0.991[b] | 0.991 |
| (46 – hb) | 4.17 | 3.99 |

Missing values indicate ILS not characterized on that day. [a]After realigning this instrument the ME was as high as 0.994 [b]As reported by Chen et al. (2016)



Table 3. EM27/SUN to Caltech TCCON Biases.

|  | Caltech, Jan | LANL, Jan | Harvard 1 | Harvard 2 | weighted %bias |
|---|---|---|---|---|---|
| n | 285 | 241 | 187 | 164 | |
| n days | 12 | 12 | 9 | 9 | |
| $X_{CO2}$ | 0.9999 (*0.16*) | 1.0006 (*0.14*) | 1.0009 (*0.15*) | 0.9998 (*0.15*) | 0.03 |
| $X_{CH4}$ | 1.0069 (*0.19*) | 1.0066 (*0.20*) | 1.0103 (*0.14*) | 1.0066 (*0.14*) | 0.75 |
| $X_{H2O}$ | 0.9840 (*1.27*) | 0.9791 (*1.44*) | 0.9886 (*1.12*) | 0.9791 (*1.01*) | -1.73 |
| $X_{CO}$ | 0.9988 (*2.30*) | | | | -0.12 |
| $X_{N2O}$ | 1.0243 (*0.42*) | | | | 2.43 |

Italicized values in parenthesis are percent standard deviations as compared to the TCCON over the

dataset.





Table 4. Meteorological sensitivity tests on EM27/SUN retrievals.

| Error | $X_{CO2}$ | | $X_{CH4}$ | | $X_{CO}$ | | $X_{N2O}$ | |
|---|---|---|---|---|---|---|---|---|
| | Offset | Daily | Offset | Daily | Offset | Daily | Offset | Daily |
| +1 hPa surf | 0.032 | 0.004 | 0.036 | 0.010 | 0.10 | 0.14 | 0.06 | 0.18 |
| +10 K (surf–700 hPa) | 0.257 | 0.076 | -0.006 | 0.036 | 10.1 | 1.2 | 0.53 | 0.23 |

Errors expressed as percentages. Daily is the median of the daily standard deviations, $Md(\sigma_{daily})$.



Table 5. Perturbations used in uncertainty budget

| perturbation | magnitude |
| --- | --- |
| ap[a] volume mixing ratio (VMR) | down shift by 1 km[b] |
| ap temperature | +1 K all altitudes |
| ap pressure | +1 hPa all altitudes |
| Pointing offset (po) | increased by 0.05° |
| Surface pressure | +1 hPa |
| Calculated Observer-Sun Doppler Stretch (OSDS) | +2 ppm |
| Field of view (FOV) | +7% |

See also Fig. 10. [a]ap=a priori [b]ap VMRs were shifted independently. For $X_{H2O}$ and $X_{HDO}$ concentrations were decreased by 50% at all levels.



Table 6. Tests for assessing biases and sensitivities of solar viewing, remote sensing instruments.

| Assessment | Test/Observation | Type | Accepted Correction[a] | Root cause | Similar instr. effect | EM27/SUN test |
|---|---|---|---|---|---|---|
| Incoming radiation attenuation effect | Gray filter after solar tracker & before interferometer | M | Recom'd replace detector. Alt. emprical | Detector non-linearity | Consistent for same detectors | § 4.4 |
| ILS | Measure with low-p gas cell (preferred), stable laser, or ambient air (least recom'd) | M | Retrievals with non-ideal ILS | Instrument misalignment. In-built | Potentially large differences | (Gisi et al., 2012; Chen et al., 2014; Frey et al., 2015) § 4.1 (measured), 5.3, 5.4 (impacts) |
| | Adjust FOV (if ILS is measured but not accounted for in retrieval) | RIA | Not recom'd | | | |
| Ghost to parent ratio | Use blackbody source & narrow band filter post interferometer | M | | Laser missampling | Likely similar, potentially large diffs | (Gisi et al., 2012; Frey et al., 2015) §4.3 |
| Ghost-effects | Measurements with & without ghost correction (e.g. XSM, or ifg resampling before FFT)[b] | M or RIA | Recom'd interpol. during acq or post resampling | Laser missampling | Likely similar, potentially large diffs | §4.3 |
| Frequency shifts | Changes or large 0 offset | O & RIA | Input spectral spacing | Improper laser wavenumber, misalignment of laser or NIR beam | Shifts differ, effect similar | §4.2 |
| Solar gas stretch | Changes or large 0 offset | O & RIA | OSDS | Poor spectral fits of solar lines. SE or res. | Similar for same detector & res. | §6 |
| Spectral fitting windows | Width, locations | RIA | | Instrument resolution requires adaptation | Same for similar res. (widths) & detector (locations) | (Gisi et al., 2012) §5.7 ($H_2O$) §5.3 (discussion) |
| Averaging kernels | Used when comparing with a different instrument type | O | Rodgers and Connor (2003) and prior info. | Diff. sensitivity at atmos. layers from differing resolutions[b] & VG | Same for similar res., microwindows & VG | §5.1 |
| SZA artifacts | Multi-day measurements in clean location | O | Empirical[a] (Wunch et al., 2011b) | ILS, or SE | See ILS entry | (Frey et al., 2015; Parker et al. 2016) |



Table 6. Continued. Tests for assessing biases and sensitivities of solar viewing, remote sensing instruments.

| Assessment | Test/Observation | Type | Accepted Correction[a] | Root cause | Similar instr. effect | EM27/SUN test |
|---|---|---|---|---|---|---|
| Long-term artifacts | Preferred co-location with accepted measurements (e.g. TCCON) | O | | Various (e.g. instrument settling, changing alignment, other) | May widely differ | Herein – for extended InGaAs only |
| Region/ zone dependence | Co-location with spatially distributed accepted measurements | O/M | | A priori insufficiencies. | Likely similar | (Parker et al., 2016) |
| Surface pressure effects | Manually adjust pressure inputs. | RIA | Accurate barometer pres. calibr. | Poor calculation of $O_2$ column, directly or by poor fitting | Similar effects for similar resolutions | §6 |
| pTz & $H_2O$ model profile sensitivity | Adjust modeled meteorological profiles | RIA | Improve met. profiles | Non-representative pTz+$H_2O$ profile | Similar effects for similar resolutions | §6 |
| A priori VMR surface sensitivity | Adjust a priori VMR near surface | RIA | Improve a prioris. Reduce effect with AKs. | Non-representative VMR profile (e.g. polluted mixed layer) | Similar effects for similar res. & true VMR profile | (Parker et al., 2016) |
| Opt. avg. time | Allan type plot (e.g. Chen et al., 2016) | O | empirical | SNR & true atmospheric variation | Depends on SNR & location | (Chen et al., 2016) §5 |
| Resolution effects | Truncate high resolution ifg | RIA | Apply offset | Inst. res. | Similar for all solar viewing insts. | (Gisi et al., 2012; Petri et al., 2012) §5.3–§5.6 |
| Uncertainty budget for current fitting algorithm | Various, test on each new algorithm (Wunch et al., 2015) | RIA | Informative | Various | Similar effects for similar resolutions | §6 |

M = measurement (setups/adjustments required before acquisition), RIA = retrieval input adjustment (post data acquisition, pre-retrieval), O = observation post retrieval (may require prior planning of locations of measurements or longer-term measurements), SE = spectroscopy errors, VG = viewing geometry, res = resolution. [a]Though empirical corrections are occasionally accepted, it is always recommended to correct the underlying problem(s) if possible. [b]I2S can provide ifg resampling if two detectors are on instrument. Note the preferred correction is always of the root cause.

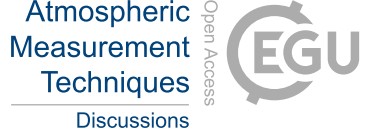



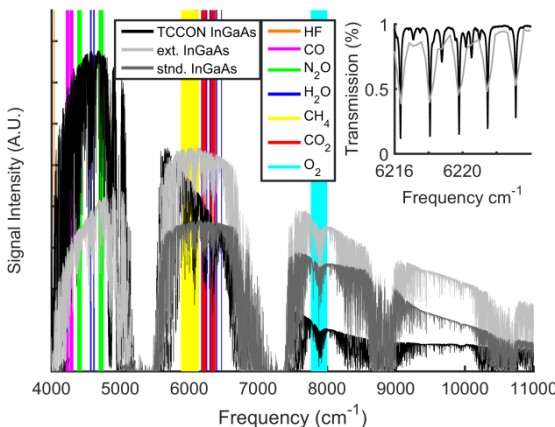

Figure 1. Example of scaled spectra from 3 different detector types, with retrieval windows highlighted. The spectrum from the EM27/SUN extended InGaAs detector was scaled 10× more than the spectrum from the standard InGaAs detector.



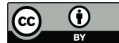

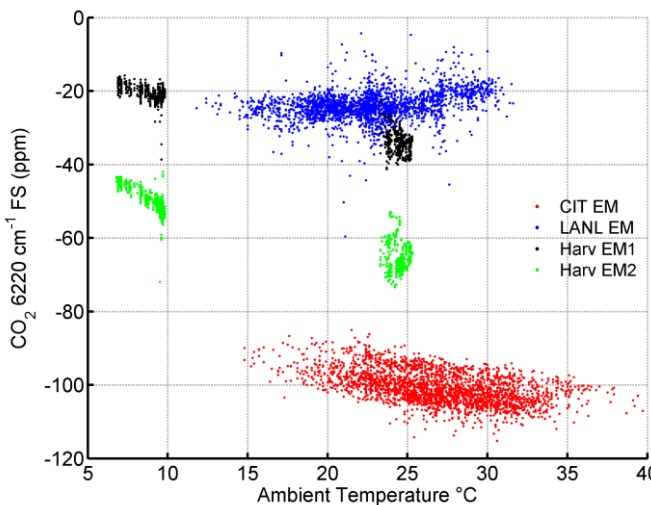

Figure 2. Frequency shifts (FS) of all 4 instruments vary with temperature because the lasers are not frequency stabilized. Shown here are FS for the $CO_2$ 6220 cm$^{-1}$ window. FS of the Caltech (CIT) instrument are far from zero, so an empirical correction is made to correct the sample spacing number. Only every 300th CIT point and every 20th LANL point is plotted for clarity. Harvard EM1 and EM2 are also referred to as ha and hb respectively by Chen et al. (2016).





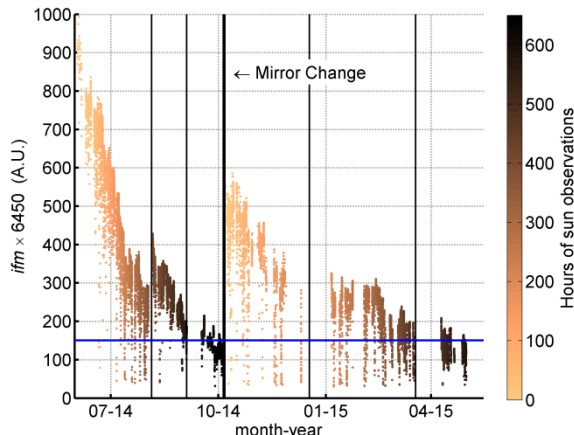

Figure 3. Interferograms from EM27/SUN instruments are negative, with the most negative values at ZPD and saturation occurring at -1. Here the interferogram maximums (*ifm*) were normalized so the maximum is 1000 and are plotted with time showing the loss of signal. Only every 50th point is plotted for clarity. Mirror cleaning (thin black lines) helped restore some signal, but never to original values. In blue is the 150 AU line.





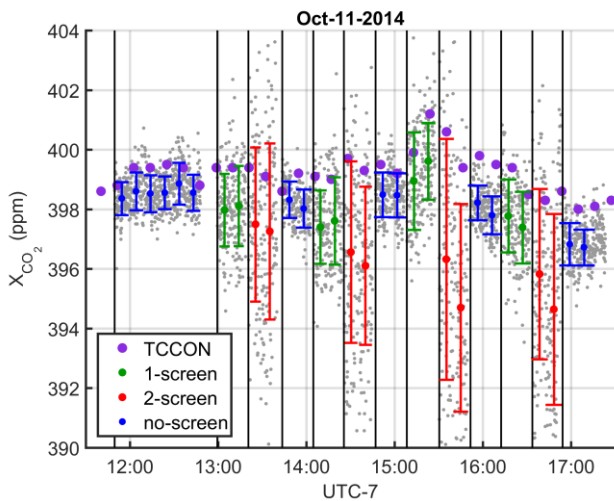

Figure 4. $X_{CO2}$ retrievals Oct. 11, 2014 when mesh screens were repeatedly moved in front of and away from the EM27/SUN (with extended InGaAs detector) entrance window. Gray points are all EM27/SUN measurements. Large points are 10-minute averages. Error bars are 1σ. This test was performed a few days after the mirrors were replaced.





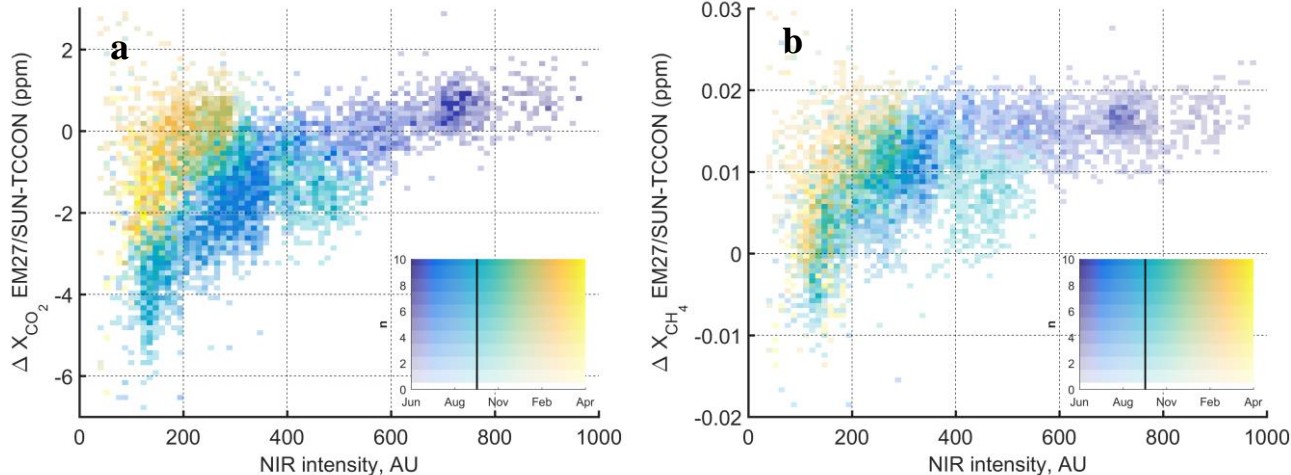

Figure 5. a. The $X_{CO2}$ retrieved from the EM27/SUN compared to TCCON decreased with signal intensity for the first set of mirrors. In October the mirrors were changed, which caused the retrieved $X_{CO2}$ to increase. The inset is the legend for the average date and number of points in the histogram bins. b. $X_{CH4}$ retrieved from the EM27/SUN compared with TCCON.





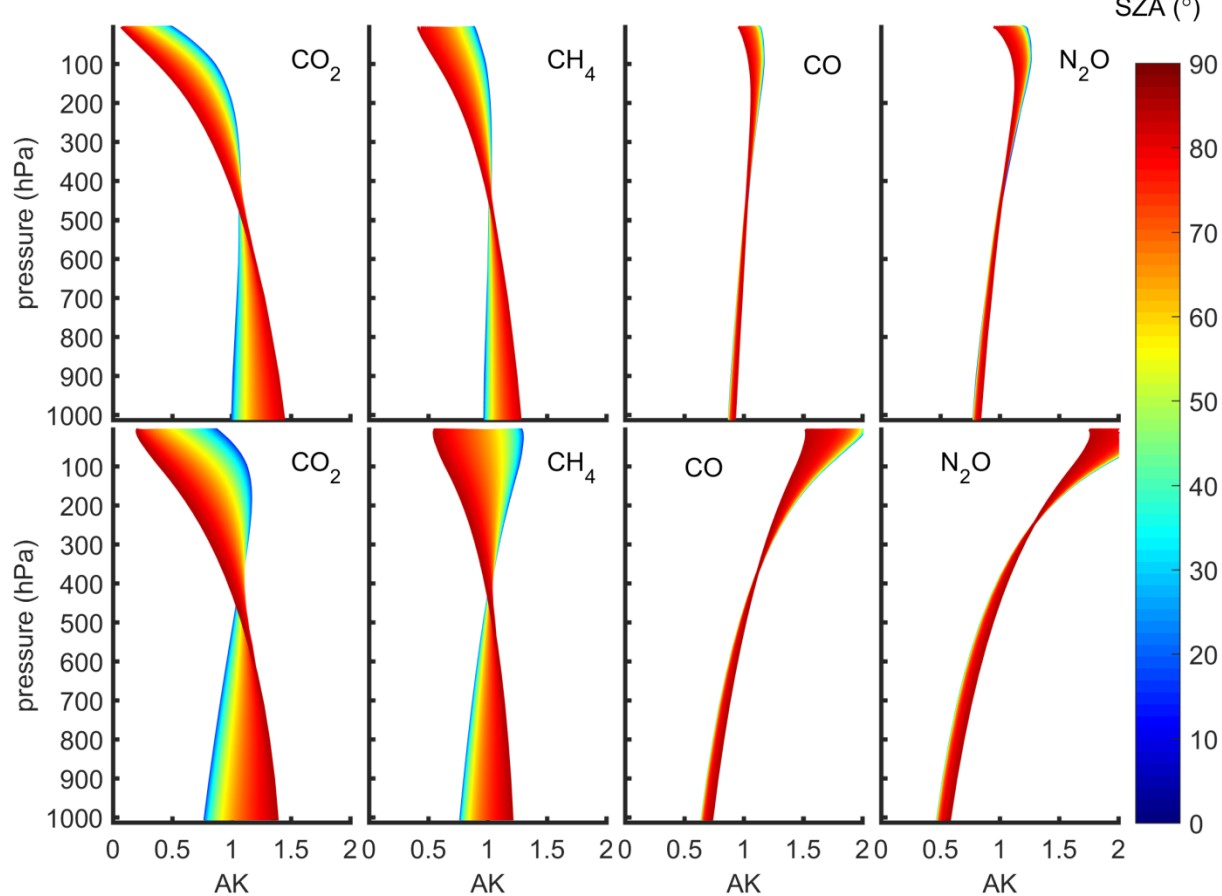

Figure 6. Top row: Averaging kernels from the Caltech EM27/SUN instrument. Bottom row: Averaging kernels from the TCCON.





Figure 7. Full time series of EM27/SUN measurements as compared to TCCON from June 2014 to May 2015. Thin vertical grey lines represent mirror cleanings. The thick line represents the mirror change. To the right are TCCON means over the time to get a sense of percent deviations.





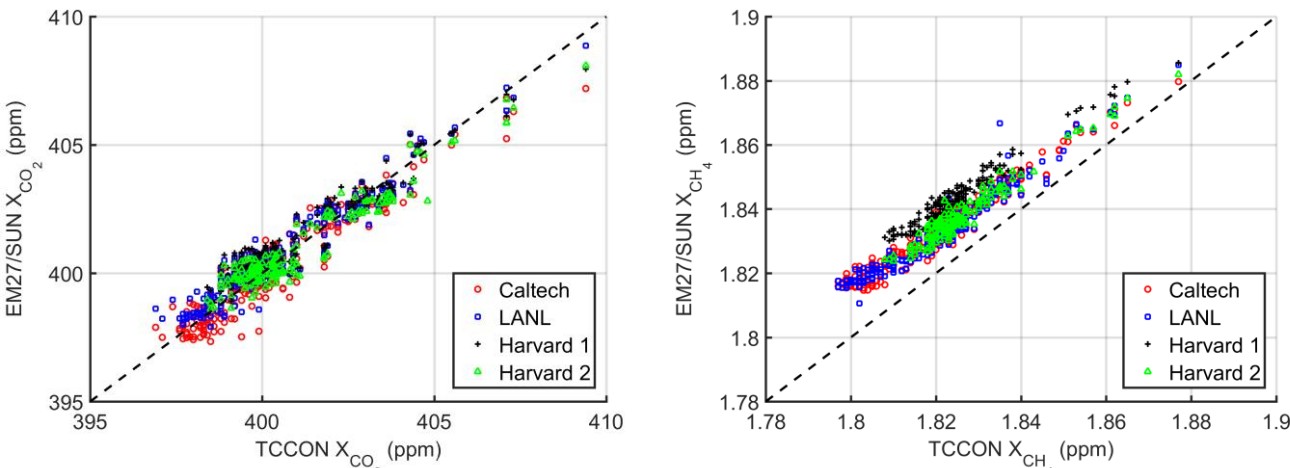

Figure 8. Retrieved EM27/SUN measurements (10 minute averaging) as compared to the TCCON from January 2015. This provides a visual representation of the data—offset and scatter of data between $X_{gas}$ from different instrument types—in Table 3. Black dashed line is 1:1 line.





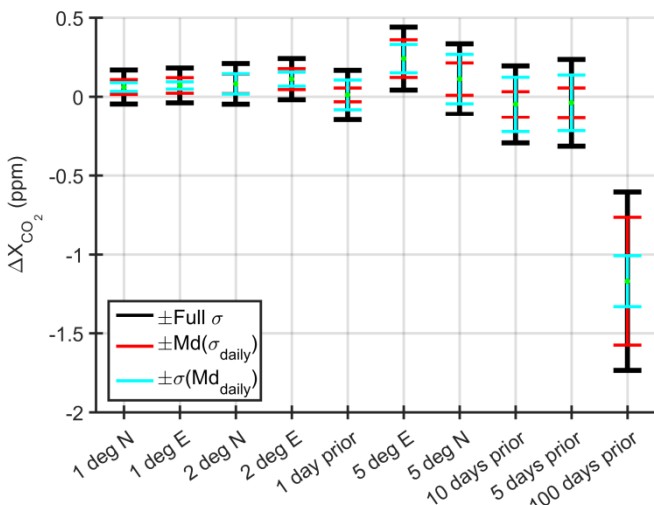

Figure 9. Standard deviations and biases from using wrong model pTz and $H_2O$ profiles as compared to using the standard option for time and location. Tests are in order of increasing full σ. Red represents intraday variability. Cyan represents interday variability.



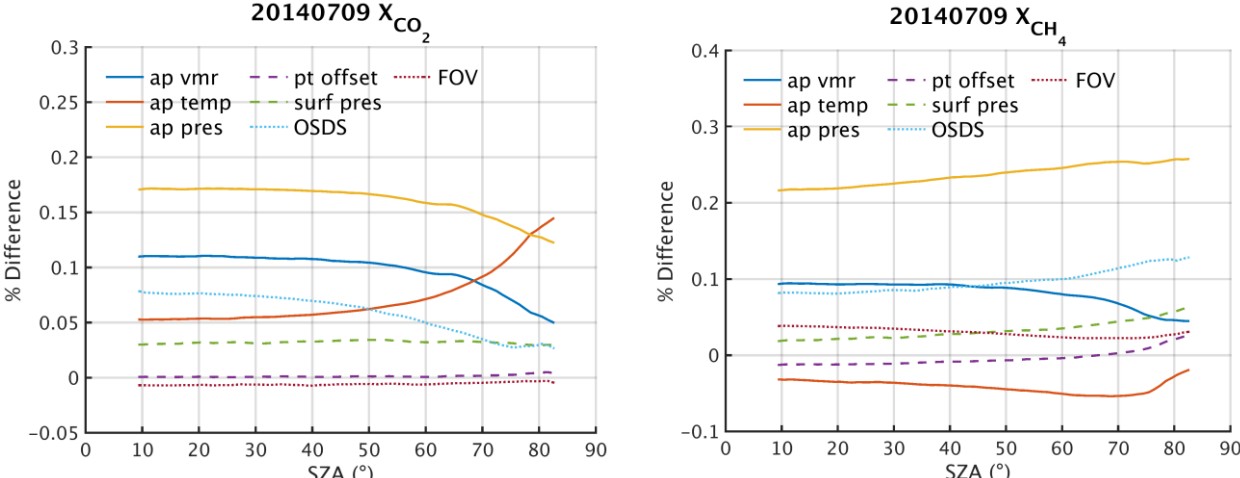

Figure 10. Uncertainty budget for EM27/SUN instruments using GGG2014. See Table 5 for magnitudes of perturbations.

