# Peer review of "Assessment of errors and biases in retrievals of $X_{\rm CO2}$ , $X_{\rm CH4}$ , $X_{\rm CO}$ , and $X_{\rm N2O}$ from a 0.5 cm-1 resolution solar viewing spectrometer"

_Atmospheric Measurement Techniques, 2016_

## Referee Comment (RC1) · Anonymous Referee #1 · 1 Apr 2016

This paper provides a valuable overview of the performance of EM27/SUN spectrometers when used for atmospheric column measurements of greenhouse gases (GHGs), compared with the standard set by TCCON measurements. The authors rightly identify the need for an expansion of the current ground based network of column GHG observations, which becomes more feasible with cheaper, more portable instrumentation such as the EM27/SUN. The primary value of this paper is in the detailed description of a methodology for comparing performance of the EM27/SUNs with TCCON, which will enable any users of EM27/SUNs (or similar ground based solar-viewing spectrometers) to assess the stability and precision of their instrument compared with a TCCON site in a manner consistent with that applied to other instruments, prior to deployment

in the field. The paper also identifies and thoroughly addresses the limitations of the EM27/SUN when used for long term monitoring, namely the non-linear response of the extended range InGaAs detector (necessitating the use of a standard spectral range InGaAs) and the impact of mirror degradation on long term stability when mirrors are exposed outside for long periods of time. I think that these results will prove to be very useful for the EM27/SUN user community in particular, and for those involved in ground based direct solar spectral measurements of the atmosphere in general.

I have a couple of specific suggestions which I think might improve the paper. Firstly, the paper alludes to quality control filters at the end of Section 2.2 which are applied to the measurements prior to the analysis. It would be useful to describe the filters used here, or at least refer to a previous publication, so that users of other EM27/SUN instruments can directly compare their performance using the analysis described here with that of the instruments studied in the paper. Similarly, I would be interested if possible to see a reference for the Bruker interpolated sampling routine mentioned briefly in Section 4.3 – I think this would be of use to those using non-Bruker spectrometers when processing their raw interferograms, prior to the retrieval stage. Overall, though, a very impressive level of detail is used to describe each part of the data-processing chain and the subsequent analysis, so in my opinion the paper is already of a high enough standard to be published.

Finally, I would like to suggest a few technical corrections:

Page 2 Line 5: Replace GHG with GHGs; Page 2 Line 8: Define acronym 'GOSAT'; Page 4 Line 17: ... spectral regions where individual gases...; Page 8 Line 24: 'cn' and 'ha' abbreviations for the instrument names are defined in the caption for Table 2, but not in the main text.

---

## Referee Comment (RC2) · M.K. Sha (Referee) · 4 Apr 2016

General comments:

This manuscript presents a study of several portable low resolution (0.5 cm-1) solar viewing FTIR spectrometers. These spectrometers are used to measure column averaged dry air mole fractions of greenhouse gases (GHGs) from ground based stations. Four such spectrometers from three research laboratories (Caltech, LANL and Harvard) are used in this study; with several results described for the Caltech EM27/SUN spectrometer. The Caltech spectrometer used in this study is a Bruker EM27/SUN spectrometer in a modified form using an extended InGaAs detector covering a spectral range of 4000 – 11000 cm-1 at first and later in its standard configuration with the

standard InGaAs detector covering a spectral range of 5500 – 12000 cm-1. The other EM27/SUN spectrometers have used a standard InGaAs detector which can measure CO2, CH4, O2 and H2O. The additional spectral range from the extended detector made the retrieval of CO and N2O possible. This is one of the key aspects of the manuscript. Furthermore, the measurements are performed right next to a high resolution solar viewing spectrometer from the Total Carbon Column Observing Network (TCCON). The EM27/SUN data is compared with respect to the TCCON data. The manuscript gives an overview of the assessment of errors and biases in the retrieval of XCO2, XCH4, XCO and XN2O. It identifies very nicely the non-linearity behavior of the extended InGaAs detector while the detector has been used in combination with a low resolution spectrometer. It gives a clear message that using the extended InGaAs detector with its full spectral range is not recommended for low-resolution spectrometers. It also identifies the effect of mirror degradation on the retrieval of the GHGs when the mirrors are exposed outside for long period of time. It also describes in detail the different steps of the data processing chain. The retrieval software suite EGI, which has been used in this study, is offered for open use to any potential user. Finally the manuscript proposes a list of tests to be performed for assessing biases and sensitivities of solar viewing remote sensing instruments. The paper gives added value to the currently existing know-how of the ground based solar absorption spectrometer community. The community will benefit from the lessons learned while using the extended InGaAs detector and its non-linearity issues in relation to the low resolution spectrometers. Therefore I recommend it for the AMT publication with some minor additions as outlined below in the specific and technical comments.

Specific comments:

I would appreciate if you could specify the conditions of your quality control filters which are used for the selection of ifms used for this study.

The long term stability of the Caltech EM27/SUN spectrometer was tested with the extended InGaAs detector which has non-linear characteristics. The data show a strong

drift in the XCO2 and XCH4 retrievals which is not so evident in the XCO and XN2O due to frequency and signal strength dependent non-linearity effects. I suppose with a proper characterization of the detector non-linearity it may be possible to understand the drift. Furthermore, I would like to mention that this does not prove that the EM27/SUN in its standard configuration (with InGaAs detector in the spectral range 5500 – 12000 cm-1) may also show long term drifts in the retrieved values of GHGs.

The author claims that it is a first time presentation of the retrieval results for XCO and XN2O. While this is true for XN2O, I would like to point out here that there has already been a publication on XCO observations using EM27/SUN by F. Hase et al. (doi:10.5194/amt-2015-403, 2016). The author should acknowledge this work and include it as a reference in this paper.

Figure 1 shows the selection of the spectral windows used for the retrieval. However, the reader has no feeling of the spectral fits for the respective gases in different micro-windows. Therefore, I would include the residual of the spectral fits for the retrieved gases for a better understanding.

Both TCCON and EM27/SUN spectrometers at Caltech use protected gold coated mirrors. However, only the latter shows a strong degradation of the mirror quality for the measurement time period. Can you please comment on the cause?

Page 18 Line 6: it says that "The non-linearity of the detector has a less pronounced effect on XCO and XN2O retrievals . . ." – Can you please spare some words on why (may be include a figure)?

Page 21 Line 18: "Our experience also suggests that the extended InGaAs detector is incompatible with precise XCO2 and XCH4 retrievals". This is a very general statement which is not necessarily true always. The author himself points out earlier that the use of a band-pass filter will be needed to operate the extended InGaAs detector in the linearity range and provide high quality measurements of CO, CO2 and CH4. The non-precise XCO2 and XCH4 retrieval was as a result of the configuration used for this

study. I would reformulate this sentence accordingly.

Figure 3: How is the ifm maximum calculated? Do you do any zero-filling? What is the reason for the intermediate increase in the ifm value (e.g. for abscissa values in-between the start and 07-14)

Technical comments:

Page 2 Line 19: I would include the formula for Xgas here.

Page3 Line 22: I would restructure the sentence as . . . The main goal of this work is to quantitatively evaluate the robustness of EM27/SUN retrievals over a long period of time.

Page 4 Line 3: . . . and data acquisition technique (or process)

Page 4 Line 4: . . . we describe the inherent properties

Page 4 Line 10: 2.1. TCCON IFS 125HR spectrometer Please replace 125 HR to IFS 125HR in the whole manuscript

Page 4 Line 13: InGaAs (Indium Gallium Arsenide)

Page 4 Line 16: I would modify "individual gases retrieved are highlighted"

Page 4 Line 18: I would include here the definition of the factor 0.2095

Page 5 Line 5: I would replace "Most use" by "The standard EM27/SUN spectrometer uses"

Page 5 Line 9: All EM27/SUN spectrometers use here . . . please give details here

Page 5 Line 22: using a medium Norton-Beer apodization with those using no special apodization.

Page 7 Line 3: the retrieval algorithm used by the TCCON

Page 18 Line 15: I would remove $\sim$ from "to EM27/SUN ($\sim$0.5"

---

## Author Comment (AC1)

**Responses to Interactive comments on: "Assessment of errors and biases in retrievals of $X_{CO2}$, $X_{CH4}$, $X_{CO}$, and $X_{N2O}$ from a 0.5 cm$^{-1}$ resolution solar viewing spectrometer" by J. K. Hedelius et al.**

We thank the referees for reviewing the manuscript and for their dedication to help improve this manuscript. Their comments are copied below (in italics) along with our responses.

**Anonymous Referee #1**

1.1) *Firstly, the paper alludes to quality control filters at the end of Section 2.2 … It would be useful to describe the filters used here.*

We have included the following sentences:

"Our QCFs were conservative and required: signal > 30 (§4.4), solar zenith angle (SZA) < 82°, 370 ppm < $X_{CO2}$ < 430 ppm, $X_{CO2,error}$ < 5 ppm, $X_{CO,error}$ < 20 ppb and $X_{CH4,error}$ < 0.1 ppm. Other users may consider stricter QCFs."

1.2) *I would be interested if possible to see a reference for the Bruker interpolated sampling routine mentioned briefly in Section 4.3 – I think this would be of use to those using non-Bruker spectrometers when processing their raw interferograms, prior to the retrieval stage.*

We now include another reference describing the laser sampling error.

"However, if the laser sampling is asymmetric—for example from a faulty electronics board—aliasing can still occur, folded across the half laser frequency (Messerschmidt et al., 2010)."

We also include a reference discussing the Bruker interpolated sampling.

"In EM27/SUN instruments the laser sampling error (LSE) can be minimized as data are collected by employing the interpolated sampling option provided by Bruker[TM]. This resampling mode uses only the rising edge of the laser interferogram and assumes constant velocity in between the rising edges to interpolate the sampling (Gisi, 2014)."

References:

Gisi, M. EM27/SUN, in: Annual Joint NDACC-IRWG & TCCON Meeting, Bad Sulza,
Germany, May 12–14, 2014.
http://www.acom.ucar.edu/irwg/IRWG_2014_presentations/Wednesday_PM/Gisi_Bruker_EN27
.pdf

Messerschmidt, J., Macatangay, R., Notholt, J., Petri, C., Warneke, T. and Weinzierl, C.: Side by
side measurements of CO2 by ground-based Fourier transform spectrometry (FTS), Tellus, Ser.
B Chem. Phys. Meteorol., 62(5), 749–758, doi:10.1111/j.1600-0889.2010.00491.x, 2010.

1.3) *Finally, I would like to suggest a few technical corrections: …*

Thank you. These were all changed as suggested.

**Referee: Dr. M. K. Sha**

2.1) *I would appreciate if you could specify the conditions of your quality control filters which*

*are used for the selection of ifms used for this study.*

Please see response 1.1.

2.2) *The long term stability of the Caltech EM27/SUN spectrometer was tested with the*

*extended InGaAs detector which has non-linear characteristics. The data show a strong drift*

*in the XCO2 and XCH4 retrievals which is not so evident in the XCO and XN2O due to*

*frequency and signal strength dependent non-linearity effects. I suppose with a proper*

*characterization of the detector non-linearity it may be possible to understand the drift.*

*Furthermore, I would like to mention that this does not prove that the EM27/SUN in its*

*standard configuration (with InGaAs detector in the spectral range 5500 – 12000 cm-1)*

*may also show long term drifts in the retrieved values of GHGs.*

Indeed the drift due to detector non-linearity characteristics is the largest (e.g. June-Sept 2014 in Fig. 8, former Fig. 7), though it appears to not be the only reason (e.g. Oct-Nov 2014). We made it clearer that additional drifts are noted that are not signal related by adding the following in §6.

"Some of these errors may partially account for the unexplained long-term drifts we noted compared to TCCON that are unrelated to signal (e.g. Fig. 8, Oct–Nov 2014)."

In §4.4 we mention that the detector response could be characterized to help understand the non-linearity. The reviewer made a good point that drifts noted in measurements made using the extended InGaAs detector do not prove there are drifts in retrievals from measurements using the standard InGaAs detector. We also want to make the point though that a lack of drift reported in former literature over short times does not imply that measurements will not drift over longer times. We note though that apparent drifts may arise from how we make our comparison or could be corrected by an updated retrieval algorithm. However, these would need to be considered anyways if EM27/SUN and TCCON data are to successfully be assimilated into the same dataset. A paragraph at the end of §6 was added discussing this.

"These long-term drifts may or may not affect instruments employing the standard InGaAs detector and may be eliminated by future retrieval updates. They may also arise in part from how the comparison was made, e.g. the assumptions to derive A4 may not be valid for $CH_4$ and $N_2O$.  As a follow-up study, brief 5–6 day comparisons using a standard InGaAs detector were made for the months of August, September, and November 2015. Scaling factors varied from 0.99905 to 1.00001 for $X_{CO2}$ and from 1.01228 to 1.00893 for $X_{CH4}$, with larger day-to-day variability. Long-term (a year or more) comparisons of these instruments employing the standard-InGaAs detector are needed before claims of long-term

| 92 | accuracy can be made or the full magnitude of drift can be quantized. Errors that could lead |
| 93 | to drifts likely would be correlated amongst all EM27/SUN instruments so the comparison |
| 94 | would need to be against a standard such as the TCCON.  Future studies may also benefit |
| 95 | from comparing results using different retrieval algorithms, as the magnitude of errors that |
| 96 | may lead to drifts in $X_{gas}$ may vary among algorithms. Meanwhile, operators have already |
| 97 | found many purposeful ways to use these instruments that require only short-term (about 1 |
| 98 | month) precision that EM27/SUN instruments using the standard detector provide without |
| 99 | any assumptions about precision for longer time periods (for example  Hase et al., 2015; |
| 100 | Chen et al., 2016; Viatte et al., 2016)." |
| 101 | |
| 102 | 2.3) *The author claims that it is a first time presentation of the retrieval results for XCO and* |
| 103 | *XN2O. While this is true for XN2O, I would like to point out here that there has already been* |
| 104 | *a publication on XCO observations using EM27/SUN by F. Hase et al. (doi:10.5194/amt-* |
| 105 | *2015-403, 2016). The author should acknowledge this work and include it as a reference in* |
| 106 | *this paper.* |
| 107 | |
| 108 | The Hase et al. (2016) paper appeared only briefly before this paper was submitted, but we |
| 109 | are happy to cite it. This paper is the first to describe $X_{CO}$ measurements from a |
| 110 | non-prototype EM27/SUN instrument in the form which Bruker sold it. The wording |
| 111 | throughout has been modified to only state that we present $X_{N2O}$ and $X_{CO}$ retrievals. We |
| 112 | have added the following to §5.5: |
| 113 | |
| 114 | "$X_{N2O}$ and $X_{CO}$ were also measured using an EM27/SUN spectrometer in this study. Hase et |
| 115 | al. (2016) have also reported on $X_{CO}$ measurements using an EM27/SUN modified to |
| 116 | include a second InGaAs detector with optical filters." |
| 117 | |
| 118 | 2.4) *I would include the residual of the spectral fits for the retrieved gases for a better* |
| 119 | *understanding.* |
| 120 | |

Now included as Fig. 7 for 9 of the different retrieval windows. We note these fits may not necessarily be representative of all spectra. We also now mention the inclusion of 11

extended-band detector benchmark interferograms in EGI. These benchmark interferograms were acquired under a variety of atmospheric conditions, but fitting these may not be representative of fits using the standard configuration. We have added the following to §5.2.

"Examples of spectral fits from several of the retrieval windows are shown in Fig. 7 for a single spectrum. These are not necessarily representative of the all conditions under which the 800,000 spectra were acquired. The residuals are larger than those reported by Gisi et al.

(2012) and Frey et al. (2015) because of the lower SNR from spectra recorded using the extended InGaAs detector."

2.5) *Both TCCON and EM27/SUN spectrometers at Caltech use protected gold coated*

*mirrors. However, only the latter shows a strong degradation of the mirror quality for the*

*measurement time period. Can you please comment on the cause?*

As a small clarification, the solar tracking mirrors for TCCON at Caltech are aluminum coated glass. There are an additional 3 mirrors that are gold that direct the light into the IFS 125HR, but they are 20 m away from the solar tracking mirrors. However, 2 TCCON sites at JPL (<10

km away) used gold coated mirrors outdoors, so we comment on them.

"The lack of degradation on the third external mirror and the JPL TCCON mirrors is likely due to differences in how the mirrors were manufactured including how the gold is applied to the substrate and the coatings used."

2.6) *Page 18 Line 6: it says that "The non-linearity of the detector has a less pronounced effect*

*on XCO and XN2O retrievals : : :" – Can you please spare some words on why (may be*

*include a figure)?*

We now point the reader to Fig. 8 (former Fig. 7) in that sentence. We feel the following two sentences are our best explanations of why the non-linearity effect is not noted for XCO

and XN2O.

"$X_{CO}$ and $X_{N2O}$ also have poorer precision than $X_{CO2}$ and $X_{CH4}$ so any non-linearity effect could be less than the noise. The 4200–4800 cm$^{-1}$ spectral region is also affected differently from the non-linearity than the 5000–7000 cm$^{-1}$ region where column $CH_4$ and $CO_2$ are retrieved from; the continuum levels changed more for the latter region. This may also explain in part why there is no noticeable change in $X_{CO}$ and $X_{N2O}$ with signal."

2.7) *Page 21 Line 18: "Our experience also suggests that the extended InGaAs detector is*

*incompatible with precise XCO2 and XCH4 retrievals". This is a very general statement*

*which is not necessarily true always. The author himself points out earlier that the use of a*

*band-pass filter will be needed to operate the extended InGaAs detector in the linearity*

*range and provide high quality measurements of CO, CO2 and CH4. The non-precise XCO2*

*and XCH4 retrieval was as a result of the configuration used for this study. I would*

*reformulate this sentence accordingly.*

Thank you for noting this. We agree the statement is too general and have modified it to now read:

"Our experience also suggests that use of the extended InGaAs detector without limiting the spectral bandpass in the EM27/SUN is incompatible with $X_{CO2}$ and $X_{CH4}$ retrievals that are precise long-term."

2.8) *Figure 3: How is the ifm maximum calculated? Do you do any zero-filling? What is the*

*reason for the intermediate increase in the ifm value (e.g. for abscissa values in-between*

*the start and 07-14)*

We have clarified that the ifm maximum is based on the maximum ordinate value of the raw interferogram. This is a value provided in the ifm file headers. The I2S routine has a zero-filling factor of 2, but they do not affect this value. Typically the ifm maximum is the peak of one of the 2 side-lobes at the centerburst. We have added the following to the Fig.

3 caption:

"Here the interferogram maximums (*ifm*) refer to the maximum (least negative) ordinate values of the raw interferograms. They were normalized so the maximum is 1000 and are plotted with time showing the loss of signal. These values are affected by clouds, which are the cause for much of the scatter. They are also affected by SZA which explains some apparent intermediate increases."

The following was also added to §4.4

"Through extended tests, we noted the first two mirrors (gold on plated aluminum, with a coating) degrade over time, with an e-folding degradation time of ~90 days as is shown in

Fig. 3. Arbitrary units (AU) for signal are the maximum ordinate values of the unmodified interferograms multiplied by 6450. The AUs of signal happen to be close to the spectral

SNR—a scaling factor of 1.3 applied to the arbitrary signal has an $R^2$ of 0.63 relative to the

SNR."

2.9) Technical comments:

Thank you; these were all changed as suggested.